# Distilling Meta Knowledge on Heterogeneous Graph for Illicit Drug Trafficker Detection on Social Media

**Yiyue Qian**[1,2]**, Yiming Zhang**[1]**, Yanfang Ye**[1,2*]**, Chuxu Zhang**[3*]

[1]Department of Computer and Data Sciences, Case Western Reserve University, USA
[2]Department of Compute Science and Engineering, University of Notre Dame, USA
[3]Department of Computer Science, Brandeis University, USA
{yiyue.qian, yiming.zhang10, yanfang.ye}@case.edu, chuxuzhang@brandeis.edu

## Abstract

Driven by the considerable profits, the crime of drug trafficking (a.k.a. illicit drug trading) has co-evolved with modern technologies, e.g., social media such as Instagram has become a popular platform for marketing and selling illicit drugs. The activities of online drug trafficking are nimble and resilient, which call for novel techniques to effectively detect, disrupt, and dismantle illicit drug trades. In this paper, we propose a holistic framework named MetaHG to automatically detect illicit drug traffickers on social media (i.e., Instagram), by tackling the following two new challenges: (1) different from existing works which merely focus on analyzing post content, MetaHG is capable of jointly modeling multi-modal content and relational structured information on social media for illicit drug trafficker detection; (2) in addition, through the proposed meta-learning technique, MetaHG addresses the issue of requiring sufficient data for model training. More specifically, in our proposed MetaHG, we first build a heterogeneous graph (HG) to comprehensively characterize the complex ecosystem of drug trafficking on social media. Then, we employ a relation-based graph convolutional neural network to learn node (i.e., user) representations over the built HG, in which we introduce graph structure refinement to compensate the sparse connection among entities in the HG for more robust node representation learning. Afterwards, we propose a meta-learning algorithm for model optimization. A self-supervised module and a knowledge distillation module are further designed to exploit unlabeled data for improving the model. Extensive experiments based on the real-world data collected from Instagram demonstrate that the proposed MetaHG outperforms state-of-the-art methods.

## 1 Introduction

As the market of illicit drugs (e.g., heroin, synthetic opioids such as Fentanyl) is considerably lucrative, the crime of drug trafficking (a.k.a. illicit drug trading) has never stopped but co-evolved with the advance of modern technologies, e.g., the practice of illicit drug trade has transformed from the physical world to online platforms. It has been shown that major social media platforms, including Instagram, Twitter, Snapchat, and Facebook, have become direct-to-consumer marketing mediums for illicit drug traffickers [25]. As illustrated in Figure 1, due to the convenience and popularity, drug traffickers can easily create accounts on these platforms to advertise and sell drugs by posting code-words (e.g., street name) and images of drugs that are in stock (shown in Figure 1.(a)); they can also post their drug sale websites and utilize social media platforms for promotion (shown in Figure 1.(b)). Buyers can access illicit drugs and trade with vendors easily through these social media platforms.

---

*Corresponding authors.

35th Conference on Neural Information Processing Systems (NeurIPS 2021).

Illegal drug trading has turn into a global concern due to its catastrophic consequences on society, from violent crimes to public health (e.g., over 87,000 people died from drug overdose in 2020 in the U.S. and it broke the record with a 20% increase in a year [41]).

Despite the persistent effort by law enforcement agencies, due to the considerable profits, activities of online drug trafficking are nimble and resilient: as shown in Figure 1.(c), to avoid being banned, skilled drug traffickers (e.g., with 1,002 followers, and 118 followings) rarely post drug description or images on social media, but instead, they advertise drugs implicitly using the slang (e.g., using "KET" to refer ketamine) and leave encrypted chat tool contact information (e.g., Kik Messenger) through their comments to other users' posts. To combat online drug trafficking, there is an imminent need to develop novel techniques for effective detection of drug traffickers on social media and thus enable law enforcement for proactive interventions to disrupt and dismantle illicit drug trades.

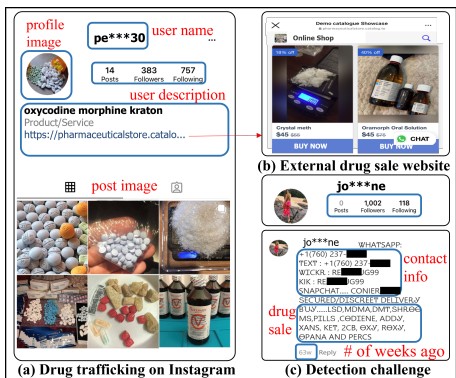

Figure 1: Showcase of drug trafficking on social media platform.

To tackle the above problem, there have been many research efforts on the investigation of online drug trafficking. Some existing works focus on analyzing the drug trafficking activities on darknet markets [46, 47, 9], while some methods have been proposed for detecting illicit drug trading activities on social media [20, 21, 22]. Most of these works usually rely on a single type of content feature (e.g., text or image) while ignoring the structured relationship information among entities on social media. For instance, Instagram monitors drug traffickers simply by filtering drug-related keyword hashtags (e.g., weed4sale) [14]. Unfortunately, as illustrated in Figure 1.(c), drug traffickers may always invent new tactics to evade detection. Those methods that fail to consider multi-modal features (e.g., both text and image) and the structural relationships among entities (e.g., users and posts) could limit the effectiveness of detecting drug trafficking activities. Additionally, most recent works [22, 23, 20] require sufficient labeled samples to train the model for the detection of drug trafficking. For example, Mackey et al. [23] collected almost 612,000 posts on Twitter and only found 1,778 drug trafficking-related posts. Obtaining labeled samples on social media is always expensive, and existing models may suffer from the constraint of few labeled data to detect drug traffickers.

To address the above challenges, in this paper, we propose a holistic framework named MetaHG (as shown in Figure 3) to automatically detect illicit drug traffickers on social media (i.e., using Instagram as a showcase). First, to overcome the shortcomings of merely analyzing a single type of content information (e.g., text or image), we propose to consider the multi-modal content features (i.e., both text and image) and introduce a heterogeneous graph (HG) to model the relationships among three types of entities (i.e., user, post, and keyword). The constructed HG is able to comprehensively characterize the complex ecosystem of drug trafficking on social media. And then we exploit relation-based graph convolutional neural networks (R-GCNs) [32] to fuse both relational information among entities and content features to obtain initial node embeddings over the built HG. Due to spare relational connections among some entities, we further introduce graph structure refinement (GSR) [24, 16] to learn more robust node representations on the HG. Furthermore, to address the few labeled data issue, we propose a meta-learning framework to transfer knowledge from training tasks and effectively adapt them to testing tasks (e.g., new types of drug traffickers with few labeled samples). More specifically, to exploit unlabeled information for better performance, our proposed model consists of two additional modules: an embedding similarity-based self-supervised learning module to augment R-GCNs for node representation refinement, and a knowledge distillation module to facilitate better model optimization. In summary, the major contributions of our work include:

- We study the illicit drug trafficker detection on social media and create a new dataset on Instagram (including text, image and relation information) for the problem, which is novel and urgent.
- To solve the problem, we develop a holistic yet novel framework (i.e., MetaHG) to jointly model both structured relations and unstructured content information for illicit drug trafficker detection, accounting for the constraints of graph sparsity and limited labeled data for model training.
- Comprehensive experiments on the real-world dataset demonstrate the outstanding performance of MetaHG by comparison with the state-of-arts methods. To the best of our knowledge, MetaHG is the first work that automatically detects drug traffickers on social media using few labeled data.

• As drug overdose deaths have continued to increase over the past decade across the country, our proposed technique will have a significant societal impact to help address this critical issue.

## 2   Related Work

In this section, we briefly introduce relevant studies in three aspects: drug trafficker detection, graph neural networks, and meta-learning.

**Drug Trafficker Detection.**   Most existing works analyze drug traffickers on darknet markets [2, 3, 46, 47, 48]. For instance, Zhang et al. [46] developed a system by leveraging user's styles to detect drug traffickers on darknet markets. As drug trafficking appears on the surface web dramatically recently, some recent works [20, 22, 23] detect illicit drug traffickers on social media platforms by analyzing the single type of text information or image information of posts. For example, Li et al. [20] used a RNN model to learn the text pattern of posts to detect drug dealers on Instagram. However, these methods seldom consider multi-modal features and the relational information among entities. Different from existing works, we detect these illicit drug traffickers on social media by incorporating structural relation and unstructured multi-modal content information.

**Graph Neural Networks.**   Most existing graph neural networks (GNNs) learn the node embedding by aggregating the features of neighborhood nodes [12, 17, 39, 44]. GCN [17] implements layer-wise propagation rule to learn the node embedding, and GAT[39] proposes to learn different attention scores for neighbors when aggregating neighborhood information. However, most of these existing models work with homogeneous graphs. Some recent models [32, 34, 43, 45] are proposed to deal with heterogeneous graphs which are more practical in reality. Inspired by GCN, Weighted-GCN [34] and R-GCNs [32] propose to learn node embedding based on multiple relational neighborhoods. Additionally, inspired by GAT, HAN [43] leverages the attention mechanism to learn the importance between nodes and meta-path simultaneously in heterogeneous graphs. Motivated by these studies, we employ R-GCNs as our base model to learn the node embedding in HG.

**Meta-Learning.**   Recent meta-learning models generally can be divided into two groups: metric-based meta-learning [4, 18, 28, 36, 40] and gradient-based meta-learning [1, 8, 10, 11, 19, 27]. For metric-based models, they implement a generalized metric and matching functions from training tasks to train the model. For instance, Matching Networks [40] learns a network that maps the labeled support set and an unlabelled example to the label in support set. Prototypical networks [36] learns a metric space in which classification can be performed by computing distances to prototype representations of each class. For gradient-based models, they employ existing tasks data to learn well initialized model parameters that can be fast updated to new tasks with few data [11] or directly implement a meta-optimizer to learn the optimization process [38]. For example, Finn et al. [10] proposed MAML, in which the parameters of the model are explicitly trained such that several gradient steps with few training data from a new task will produce good generalization performance on that task. In this paper, motivated by MAML, we introduce a meta-learning framework to address small labeled data challenge of illicit drug traffickers detection.

## 3   Preliminary

### 3.1   Problem Definition

Given a social media dataset, we build a heterogeneous graph (HG) to comprehensively depict the rich information among the dataset for learning the user embedding. Let $G = (\mathcal{V}, \mathcal{E}, \mathcal{X})$ denote a HG, where $\mathcal{V}$ is the set of different types of nodes, $\mathcal{E} \subseteq \mathcal{V} \times \mathcal{V}$ is the set of edges, and $\mathcal{X}$ is the node attributes (features) set. Specifically, there are three types of nodes (user, post, and keyword) and eight types of relations (e.g., user-reply-post) in HG (see Figure 2). A node can be regarded as any type of entities and an edge can be regarded as any type of relations between two nodes. The node attribute can be considered as the feature applied to the node. The goal is to learn the user embedding which can be fed to a classifier for drug trafficker detection. Given the features of all user nodes $X_r = (x_1, \ldots, x_N)$ ($N$: number of users) and their binary labels $Y = (y_1, \ldots, y_N)$ ($y_i = 1$ denotes drug trafficker and $y_i = 0$ represents regular user), we target learning a mapping function (detection function) $U_\phi : X_r \to Y$ (with parameter $\phi$). Unlike existing works that use sufficient

samples for model training, as the number of drug trafficker samples collected from social media is limited, we consider a more practical scenario that only few labeled data are available. Following the definition of few-shot learning [10], given the labeled data of existing different types of drug traffickers (e.g., stimulants trafficker (refer to the Supplementary Material 1.1)), we aim to build a classifier that can effectively detect drug traffickers of new types with few labeled data (e.g., opioid trafficker). Formally, the problem is defined as follows.

**Problem 1.** ***Drug Trafficker Detection on Social Media.*** *Given a social media data denoted as HG $G$ and a set of different types of drug traffickers with their features $X_r = (x_1, \ldots, x_N)$ and corresponding labels $Y = (y_1, \ldots, y_N)$ (training data), the problem is to build a machine learning model to detect illicit drug traffickers of new types that only have few labeled samples (testing data).*

### 3.2 Relational Graph Convolutional Networks

Relational graph convolutional networks (R-GCNs) [32] has been proved to be powerful for learning the node representation in HG. Different from GCN, R-GCNs considers the different types of relationships between two nodes when aggregating features from neighborhoods. Thus, we employ R-GCNs as the base model to learn user representations in HG. According to the definition of R-GCNs, The layer-wise propagation rule can be formulated as follows:

$$h_i^{l+1} = \sigma(\sum_{r \in \mathcal{R}} \sum_{j \in \mathcal{N}_i^r} \frac{1}{|N_i^r|} W_r^l h_j^l + W_0^l h_i^l), \tag{1}$$

where $h_i^{l+1}$ denotes the representation of node $i$ at $l+1$ layer, $W^l$ is learnable weight, $N_i^r$ denotes the set of neighbors of node $i$ under relation $r \in \mathcal{R}$, and $\mathcal{R}$ is the set of relation type in HG. For simplicity, we use $Z = \text{R-GCNs}(X, A, \mathcal{R})$ to denote a R-GCNs model where $Z$ is the node embedding, $X$ is the attribute feature matrix, and $A$ is the adjacency matrix based on $\mathcal{R}$.

## 4 Methodology

In this section, we present the details of our proposed model MetaHG (Figure 3). At first, we construct a HG to depict both relation and multi-modal content information on social media, and introduce graph structure refinement (GSR) to generate a refined HG for compensating the spare connection among some entities. Then we utilize R-GCNs over the learned graph to learn node embedding and augment the model by an embedding similarity based self-supervised learning module (SSL). Lastly, we design a meta-learning algorithm for model optimization, which is further enhanced by a knowledge distillation module (KD).

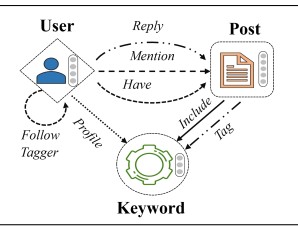

Figure 2: Network schema of HG for social media.

### 4.1 HG Construction on Social Media

To comprehensively describe illicit drug traffickers on social media, besides text and image content, we also consider the structural relation on social media. As shown in Figure 2, we build a HG with three types of entities (user, post, and keyword) and eight types of relations as well as features of each node, such that both content and relation information can be exploited simultaneously. Next, we introduce the content feature and relation information in detail.

**Content Feature.** Most users on social media post images and text simultaneously. Thus, we consider both text feature and image feature of posts and users. ***Text Feature:*** we first merge all of text information as the corpus to fine-tune the pre-trained language model *BERT* [5] and convert all of the

Table 1: Content and relation information in HG for social media.

| Content Feature | |
| --- | --- |
| User | username, profile information,is-business, # of followers/followings/posts |
| Post | text content, image information comment content, # of likes/comments |
| Keyword | keywords extracted from all text content (e.g., oxycodone, fentanyl, codine, and LSD) |

| Relation | |
| --- | --- |
| R1: user-follow/followed-user | R2: user-tagger-user |
| R3: user-reply-post | R4: user-mention-post |
| R5: user-have-post | R6: user-profile-keyword |
| R7: post-include-keyword | R8: post-tag-keyword |

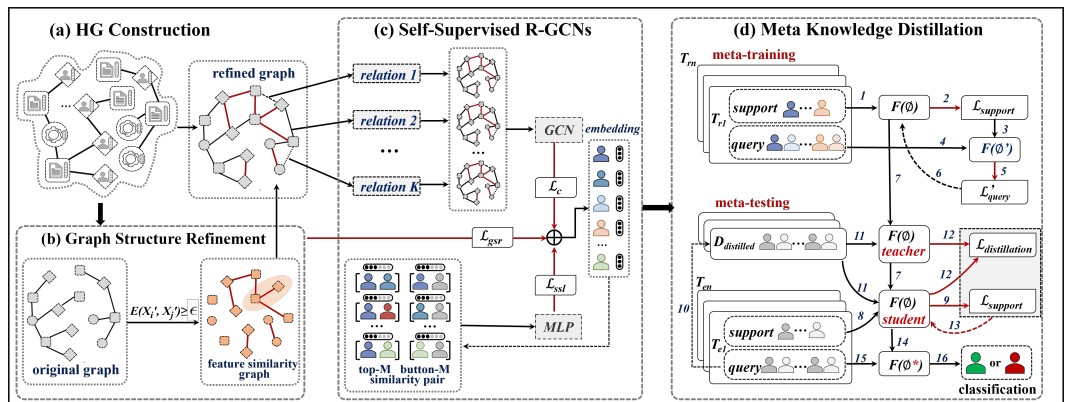

Figure 3: The overall framework of MetaHG: (a) HG construction; (b) Graph structure refinement; (c) Self-supervised R-GCNs for generating node embedding; (d) Meta-learning with knowledge distillation for drug trafficker detection.

text information for each node to a fixed-length feature vector (dimension = 200). Particularly, for keyword nodes in HG, we first extract keywords from the corpus and then select a set of illicit-oriented keywords based on word frequency. Furthermore, for each keyword node in HG, we feed the keyword to *BERT* to obtain the feature vector. ***Image Feature:*** we employ the pre-trained image model *VGG19* [35] to acquire the image feature vector (dimension = 1000) for each image and then implement *PCA* [31] to decrease the dimension from 1000 to 200. Note that the image feature for a user node is the image feature vector of the user profile image, and the image feature for a post node is the image feature vector of the post image. For all keyword nodes, the image feature is set as zero. Finally, both text and image features are concatenated as the attribute applied to each node.

**Relation.** To determine whether a user is a drug trafficker on social media, we not only consider the content-based features (text and image features), but also the complex relationships among users, posts, and keywords. To characterize the relatedness of two nodes, we consider eight kinds of relationships (R1-R8 in Table 1) as follows: *R1: user-follow/followed-user* denotes that a user is following or followed by another user; *R2: user-tagger-user* denotes that a user tags another user; *R3: user-reply-post* denotes that a user replies to a post; *R4: user-mention-post* denotes that a user mentions another user in a post; *R5: user-have-post* denotes that a post belongs to the user; *R6: user-profile-keyword* denotes that the profile description of a user contains the keyword; *R7: post-include-keyword* denotes that the post content includes a certain keyword; *R8: post-tag-keyword* denotes that the content of a post has the hashtag keyword.

## 4.2 HG Structure Refinement via Metric Learning

As some users appear to be inactive on social media, the HG constructed in the previous step is sparse in connections among some nodes. That is, the original HG is not optimal for detecting drug traffickers on social media. To address this problem, we propose to generate a better HG by graph structure refinement with metric learning (GSR). Specifically, we first calculate the similarities between node features to infer the potential connections between nodes by metric learning and obtain the feature similarity graph (FSG). Then, the original HG and FSG are combined to obtain a new and refined graph (see Figure 3.(b)). The generation process of FSG is described as below.

**Feature Similarity Graph.** Since the dimension of different types of node features is different, we first adopt a type-specific mapping layer to project all of the node features $X$ to a common feature space:

$$X_i^{'} = \sigma(X_i \cdot W_k + b_k), \tag{2}$$

where $\sigma$ is the *sigmoid* activation function, $X_i$ is the feature vector of node $v_i$, $W_k$ and $b_k$ denote the projection matrix and the bias vector for node type $k$ respectively. Afterwards, we implement metric learning on node features in the common space and obtain a FSG $\mathcal{G}'$ where the edge between node pair $v_i$ and $v_j$ is obtained by:

$$\mathcal{E}_{i,j}^{'} = \begin{cases} 1 & E(X_i^{'}, X_j^{'}) \geq \epsilon \\ 0 & \text{otherwise} \end{cases}, \tag{3}$$

where $\epsilon \in [0, 1]$ is the threshold value controlling the density of the generated FSG. $E(X_i^{'}, X_j^{'})$ is a parameterized function to calculate the similarity between two nodes:

$$E(X_i^{'}, X_j^{'}) = \mathbf{\Gamma}(W_{s1} \odot X_i^{'}, W_{s2} \odot X_j^{'}), \tag{4}$$

where $\mathbf{\Gamma}$ denotes a similarity function (*cosine* similarity in this work), $W_{s1}$ and $W_{s2}$ are the parameters which measure the importance of different feature dimensions, and $\odot$ denotes the Hadamard product. The refined graph $\hat{\mathcal{G}}^l$ is obtained by combining the original HG $\mathcal{G}$ and the generated FSG $\mathcal{G}'$. In order to make $\mathcal{G}^l$ be relatively sparse and avoid overfitting for the downstream task, we introduce a structure sparsity regularizer via $L_1$ norms:

$$\mathcal{L}_{gsr} = \left\| A^l \right\|, \tag{5}$$

where $A^l$ is the adjacency matrix of $\mathcal{G}^l$.

## 4.3  HG Representation Learning with Self-Supervised Augmentation

With the refined graph $\mathcal{G}^l$ obtained in the previous step, we implement R-GCNs to learn node (i.e., user) embedding for drug trafficker detection. In particular, we stack R-GCNs layers to generate node embedding (Equation 1). For each user node $v_i \in \mathcal{V}^c$, we feed $h_i$ into a fully-connected layer and *softmax* function to predict illegal value of drug trafficker, i.e., $\hat{Y}_i = h_i W_c$. The objective is to jointly optimize the drug trafficker detection loss (i.e., the cross-entropy loss between the given drug trafficker labels $Y$ and the predicted results $\hat{Y}$) and the GSR loss $\mathcal{L}_{gsr}$:

$$\mathcal{L} = \mathcal{L}_c + \lambda_{gsr}\mathcal{L}_{gsr} = -\sum_{i \in \mathcal{V}^c} Y_i \log(\hat{Y}_i) + \lambda_{gsr} \left\| A^l \right\|, \tag{6}$$

where $\mathcal{V}^c$ is the node set of labeled users in HG, $Y$ is the user label set, and $\lambda_{gsr}$ is a trade-off weight.

**Self-Supervised Augmentation.** Existing GNNs [17, 12, 26, 30] utilize the local neighborhood information and feature content to generate the node embedding. However, the representation learning in most GNNs is mainly dominated by local neighborhood information [15]. We wish to achieve a balance among the importance of local neighborhood and node features. Hence, we introduce a node embedding similarity based self-supervised module (SSL) using unlabeled data to enhance the HG representation learning (see Figure 3.(c)). Specifically, given a node $v_i$, we define two sets of nodes $\mathcal{V}_i^h$ and $\mathcal{V}_i^l$ having the $M$ highest similarities and $M$ lowest similarities with $v_i$ respectively. Based on the similarity (*cosine* similarity) among pairs of original node features, we can define node pairs in $\mathcal{V}_i^h$ as positive samples while node pairs in $\mathcal{V}_i^l$ as negative samples. We wish that the embedding of node pairs in $\mathcal{V}_i^h$ should be more similar while the embedding of node pairs in $\mathcal{V}_i^l$ should be less similar. Hence, in each training epoch, we concatenate the embedding of each node pair $(v_i, v_q)$ $(i \neq q)$ in $\mathcal{V}_i^h$ and $\mathcal{V}_i^l$ respectively denoted as $X_{iq}$, and assign each pair with label $Y_{iq}$. $Y_{iq} = 1$ when $v_q \in \mathcal{V}_i^h$ otherwise $Y_{iq} = 0$ when $v_q \in \mathcal{V}_i^l$. Afterwards, we feed the node pair embedding $X_{iq}$ into a fully-connected neural network, i.e., $\hat{Y}_{iq} = X_{iq}W_{ssl}$, to obtain the predicted label. Finally, the final objective of node representation learning is to minimize the joint loss including drug trafficker detection loss, the SSL loss, and the GSR loss:

$$\mathcal{L}_{joint} = \mathcal{L}_c + \lambda_{ssl}\mathcal{L}_{ssl} + \lambda_{gsr}\mathcal{L}_{gsr} = -\sum_{i \in \mathcal{V}^c} Y_i \log(\hat{Y}_i) - \lambda_{ssl} \sum_{i \in \mathcal{V}} \sum_{q \in \mathcal{V}_i^s} Y_{iq} \log(\hat{Y}_{iq}) + \lambda_{gsr} \left\| A^l \right\|, \tag{7}$$

where $\mathcal{V}_i^s = \mathcal{V}_i^h \cup \mathcal{V}_i^l$ and $\lambda_{gsr}$ is the hyper-parameters of SSL. By minimizing $\mathcal{L}_{joint}$, parameters of R-GCNs, SSL, and GSR are optimized jointly for downstream drug trafficker classification task.

## 4.4  Model Optimization with Meta-Learning

We propose to leverage the meta-learning technique [10] for model optimization since the labeled data of drug traffickers is limited. In particular, it applies the gradient-based algorithm that learns well initialized model parameters (using training tasks data) which can be quickly adapted to unseen new tasks. In this paper, given a set of tasks $\mathcal{T}$ defined as drug trafficker detection problem (binary classification) of different trafficker types, we choose three drug trafficker types with a relatively large number of samples (e.g., stimulants trafficker) as meta-training tasks and the other two types with few

---

**Algorithm 1** Training Procedure of MetaHG

---

**Require** $X$, $A$, $\mathcal{R}$, $\phi$: nodes features, adjacent matrix, node relations, and initial parameters.
   Learn a refined graph via Equation 3 and 4.
   Implement self-supervised learning to augment R-GCNs on the refined graph.
   **while** not convergae **do**
      Sample a batch of meta-training tasks $\tau$ from $\mathcal{T}$.
      **for** each $\tau$ **do**
         Sample a support set $\mathcal{S}_\tau$ and a query set $\mathcal{Q}_\tau$.
         Update the parameters $\phi'_\tau$ via Equation 8.
      **end for**
      Update the parameters $\phi$ of task-agnostic model via Equation 9.
      Sample support set $\mathcal{S}_\tau$ and query set $\mathcal{Q}_\tau$ from meta-testing tasks.
      Distill the soft knowledge from teacher model using the unlabeled data $\mathcal{Q}_\tau$ via Equation 10.
      Update the parameters of student model via optimizing Equation 13.
   **end while**
   **Return** Optimized $\phi^*$

---

samples (e.g., opioid trafficker) as meta-testing tasks. Then, we divide the data of each task $\tau \in \mathcal{T}$ into support set $\mathcal{S}_\tau$ and query set $\mathcal{Q}_\tau$. Afterwards, the classifier is first updated to task-specific model on meta-training tasks with support set $\mathcal{S}_\tau$, and is further optimized to task-agnostic model using $\mathcal{Q}_\tau$ of training tasks, which is called meta-training. After sufficient training, the learned model can further adapt to new testing tasks with few data samples in support set, which is called meta-testing. Let $\phi$ be the set of model parameters. For a certain classification task $\tau$, we begin with feeding $\mathcal{S}_\tau$ to the model and calculate the loss $\mathcal{L}_\tau$ on $\mathcal{Q}_\tau$ to update $\phi$ to $\phi'_\tau$ through gradient descent:

$$\phi'_\tau = \phi - \alpha \bigtriangledown_\phi \mathcal{L}_\tau(\phi), \tag{8}$$

where $\alpha$ is the step size for inner-level meta-training. Later, the model (with parameter $\phi$) is further updated in task-agnostic manner:

$$\phi \leftarrow \phi - \beta \bigtriangledown_\phi \sum_{\tau \in \mathcal{T}} \mathcal{L}_\tau(\phi'_\tau), \tag{9}$$

where $\beta$ is the learning rate. After sufficient training over meta-training tasks, the well initialized parameters is further adapted for prediction over meta-testing tasks.

**Meta Knowledge Distillation for Model Adaptation.** The above optimization strategy directly adapts the classifier learned from the labeled data in meta-training to new testing tasks, while ignoring the the hidden information of unlabeled data during model training. Inspired by the knowledge distillation technique [13], we develop a meta knowledge distillation module (KD) that utilizes the unlabeled data in meta-learning to transfer soft knowledge from the meta-training classifier (teacher model) to the meta-testing model (student model) (Figure 3.(d)). Mimicking teacher's prediction results enables student model to learn the secondary information that cannot be expressed by the labeled samples in meta-testing data alone. Soft knowledge from teacher model is formulated as the predicted probability of drug traffickers in meta-testing data:

$$P_l^T(Z_i, t) = \text{Softmax}(f(Z_i), t)) = \frac{\exp\left[f_l(Z_i/t)\right]}{\sum_{c=0}^{1} \exp\left[f_c(Z_i/t)\right]}, \tag{10}$$

where $Z_i$ is the embedding of node $v_i$ in $\mathcal{Q}_\tau$ for meta-testing task $\tau$, $f(Z_i)$ is the score logit that $Z_i$ achieves on label $l$, and $t$ is the temperature index to soften the peaky softmax distribution [13]. Thus, the knowledge distillation loss of teacher model and student model is defined as follows:

$$\mathcal{L}_{kd} = -t^2 \sum_{v_i \in \mathcal{Q}_\tau} \sum_{c=0}^{1} P_c^T(Z_i, t) \log(P_c^S(Z_i, t)), \tag{11}$$

where $P_c^T$ and $P_c^S$ are the predicted distributions of teacher model and student model respectively. Afterwards, we combine the KD loss $\mathcal{L}_{kd}$ on unlabeled data (query data) and the cross-entropy loss $\mathcal{L}_{ce}$ on labeled data (support set):

$$\mathcal{L}_{total} = \mathcal{L}_{ce} + \lambda_{kd}\mathcal{L}_{kd}, \tag{12}$$

where $\lambda_{kd}$ is the trade-off weight for balancing the two losses. Taking the model (with parameter $\phi$) obtained by Equation 9 as the initial model, the student model for meta-testing tasks are further updated (fine-tuned) as follows:

$$\phi^* = \phi - \alpha \bigtriangledown_\phi \mathcal{L}_{total}(\phi), \qquad (13)$$

where $\alpha$ is the learning rate. Finally, the fine-tuned student model is used to detect drug trafficker in query set of meta-testing tasks. The pseudo-code of MetaHG training procedure is shown in Alogrithm 1.

## 5 Experiments

In this section, we first create a novel dataset for the problem. Then we conduct extensive experiments to evaluate the performance of our model. Further analysis is provided to show the effectiveness of each model component, hyper-parameter sensitivity, and model stability.

### 5.1 Novel Dataset for Drug Trafficker Detection

Existing works of drug trafficker detection on social media merely rely on text or image data and most of the datasets are not publicly accessible. Thus, we create a new dataset crawled from Instagram (a very popular social media platform worldwide [42]) for the problem. Specifically, we first utilize a set of illegal drug related keywords (e.g., weed, LSD, and fentanyl) to crawl the public user profiles and the corresponding posts using the public official Instagram API [7] from Jan 2020 to Jan 2021. Note that all collected data are public and we have anonymized the personal information of public users in our dataset. To obtain the groundtruth labels, we (including 3 groups, 2 annotators in each group) spent two months collecting and manually labeling these users into six groups (regular user, stimulants trafficker, hallucinogens trafficker, opioids trafficker, hidden trafficker, and mixture trafficker) according to the criterion in [33]. In total, we obtain 8,651 users (including 3,242 drug traffickers and 5,409 regular users) and 79,705 posts. Based on the dataset, we construct a HG which has 129,894 nodes (including 8,651 user nodes, 79,705 post nodes, and 41,538 keyword nodes) and 218,039 edges of eight relations (Table 1). Details of this dataset is provided in Supplementary Material 1.1.

### 5.2 Baseline Methods

We compare our model with twenty baseline methods spanning several categories:

- **Traditional Classification Methods**. We take text features (tFeature), image features (iFeature), and the combination of them (cFeature) as the feature vector for each user respectively, and feed them to a generic 4-layer deep neural network [37] (**DNN**) (B1). We also consider two existing works [20, 22] using tFeature for drug traffickers detection (B2).
- **Meta-Learning Models**. For meta-learning baseline methods, besides **MAML** [10], we also employ two popular few-shot learning models, i.e., Matching Network (**MatchingNet**) [40] and Prototypical Network (**ProtoNet**) [36] to train a meta-learner with cFeature (B3).
- **Graph Representation Learning Models**. We take cFeature as node attribute information and apply six popular graph representation learning models to learn user embeddings in HG, i.e., **Deepwalk** [29], **metapath2vec** [6], **GCN** [17], **GAT** [39], **HAN** [43], and **R-GCNs** [32]. The learned user embeddings are further fed to a 4-layer DNN (B4) for classification. In addition, we also apply MAML to train these graph representation learning models (B5).

The details of all models are discussed in Supplementary Material 2.1.

### 5.3 Performance Comparison

Table 2 reports the performances of all models. The best performances are highlighted in bold. All baseline models are divided into five groups (B1-B5) and the shot number denotes the support data size in model training and testing. According to the table, we can conclude that (i) The combination of text and image features (cFeature) contributes larger than single text or image features for drug traffickers detection (B1). It shows that both text and image are very indispensable to describe users on social media. (ii) By comparing results in B1 and B4, we find that considering the relationships among

Table 2: Performance comparisons of all methods with different support data sizes (shot numbers).

| Setting | | 1-shot | | 5-shot | | 10-shot | | 20-shot | |
|---|---|---|---|---|---|---|---|---|---|
| Group | Model | ACC | F1 | ACC | F1 | ACC | F1 | ACC | F1 |
| B1 | tFeature+DNN [37] | 0.4786 | 0.3768 | 0.5046 | 0.3959 | 0.5215 | 0.4116 | 0.5420 | 0.4230 |
| | iFeature+DNN | 0.5253 | 0.4256 | 0.5428 | 0.4432 | 0.5564 | 0.4635 | 0.5712 | 0.4841 |
| | cFeature+DNN | 0.5468 | 0.4434 | 0.5601 | 0.4553 | 0.5758 | 0.4755 | 0.5935 | 0.4924 |
| B2 | Li *et al.* [20] | 0.4929 | 0.3942 | 0.5287 | 0.4171 | 0.5496 | 0.4280 | 0.5701 | 0.4456 |
| | Rokon *et al.* [22] | 0.5058 | 0.4037 | 0.5396 | 0.4260 | 0.5585 | 0.4376 | 0.5765 | 0.4529 |
| B3 | cFeature+ProtoNet [36] | 0.5815 | 0.5735 | 0.6156 | 0.5987 | 0.6321 | 0.6293 | 0.6637 | 0.6572 |
| | cFeature+MatchingNet [40] | 0.6058 | 0.5953 | 0.6337 | 0.6257 | 0.6551 | 0.6478 | 0.6773 | 0.6659 |
| | cFeature+MAML [10] | 0.6337 | 0.6218 | 0.6654 | 0.6525 | 0.6872 | 0.6735 | 0.6959 | 0.6985 |
| B4 | [29] Deepwalk+DNN | 0.5919 | 0.4991 | 0.6210 | 0.5251 | 0.6472 | 0.5449 | 0.6710 | 0.5605 |
| | [6] metapath2vec+DNN | 0.6257 | 0.5243 | 0.6518 | 0.5549 | 0.6739 | 0.5683 | 0.6953 | 0.5825 |
| | [17] GCN+DNN | 0.6524 | 0.5510 | 0.6853 | 0.5772 | 0.7054 | 0.5953 | 0.7291 | 0.6138 |
| | [39] GAT+DNN | 0.6650 | 0.5558 | 0.6889 | 0.5842 | 0.7129 | 0.6023 | 0.7305 | 0.6195 |
| | [43] HAN+DNN | 0.6786 | 0.5629 | 0.7047 | 0.5925 | 0.7207 | 0.6152 | 0.7421 | 0.6290 |
| | [32] R-GCNs+DNN | 0.6836 | 0.5765 | 0.7183 | 0.6042 | 0.7254 | 0.6221 | 0.7476 | 0.6446 |
| B5 | Deepwalk+MAML | 0.6962 | 0.6957 | 0.7324 | 0.7306 | 0.7559 | 0.7537 | 0.7751 | 0.7674 |
| | metapath2vec+MAML | 0.7251 | 0.7232 | 0.7622 | 0.7534 | 0.7837 | 0.7746 | 0.7995 | 0.7921 |
| | GCN+MAML | 0.7526 | 0.7407 | 0.7835 | 0.7827 | 0.8052 | 0.7924 | 0.8319 | 0.8356 |
| | GAT+MAML | 0.7578 | 0.7426 | 0.7905 | 0.7921 | 0.8124 | 0.8036 | 0.8439 | 0.8214 |
| | HAN+MAML | 0.7732 | 0.7654 | 0.8062 | 0.7959 | 0.8328 | 0.8176 | 0.8551 | 0.8327 |
| | R-GCNs+MAML | 0.7853 | 0.7727 | 0.8253 | 0.8149 | 0.8465 | 0.8352 | 0.8678 | 0.8535 |
| Ours | **MetaHG** | **0.8489** | **0.8480** | **0.8873** | **0.8758** | **0.9196** | **0.9122** | **0.9354** | **0.9311** |

entities can help improve the model performance. (iii) The performance of meta-learning models with combined features (B3) or node embeddings (B5) are much better than traditional classification models (B1 and B2), showing the effectiveness of meta-learning for solving this problem. (iv) In all cases, MetaHG significantly outperforms all baseline methods, demonstrating the strongest capability of our model for drug traffickers detection on social media.

## 5.4 Ablation Studies

Our model MetaHG integrates four crucial components, i.e., graph structure refinement (GSR), self-supervised learning in R-GCNs (SSL), meta-learning (MAML), and knowledge distillation (KD). To verify the effectiveness of each component, we conduct ablation studies by removing each of them independently. The results of different model variants are reported

Table 3: Results of different model variants.

| Model | 1-shot | | 5-shot | | 20-shot | |
|---|---|---|---|---|---|---|
| | ACC | F1 | ACC | F1 | ACC | F1 |
| **MetaHG** | **0.8489** | **0.8480** | **0.8873** | **0.8758** | **0.9354** | **0.9311** |
| − MAML (A1) | 0.7247 | 0.6125 | 0.7535 | 0.6457 | 0.7932 | 0.6851 |
| − KD (A2) | 0.7914 | 0.7922 | 0.8356 | 0.8247 | 0.8804 | 0.8689 |
| − GSR (A3) | 0.8078 | 0.8018 | 0.8457 | 0.8351 | 0.8934 | 0.8857 |
| − SSL (A4) | 0.8147 | 0.8052 | 0.8561 | 0.8473 | 0.9056 | 0.9005 |

in Table 3. We first replace MAML with a 4-layer DNN (A1), which means we implement the node embedding generated from SSL augmented R-GCNs over the refined graph. Similar to the data setting in B1 and B4, all labeled data in meta-training tasks and few-shot labeled data (support set) in meta-testing tasks are for model training, and the rest of data in meta-testing tasks (query set) are for model evaluation. We conclude that MAML has the greatest contribution to MetaHG. Then we remove KD on MAML (A2) and find the performance decrease obviously, showing the effectiveness of KD. Furthermore, we remove GSR (A3) and SSL (A4) from MetaHG respectively and find the performances of A3 and A4 drop almost 4% and 3% respectively compared with MetaHG, validating that both GSR and SSL are effective to enhance the model performance for the detection problem.

## 5.5 Hyper-parameter Sensitivity and Model Stability

To explore the hyper-parameter sensitivity, we conduct three analysis experiments w.r.t. $\epsilon$ on GSR, $\lambda_{ssl}$ on SSL, and $\lambda_{kd}$ on meta-learning. Specifically, in Figure 4.(a), we vary $\epsilon$ in Eq. 3 to control the density of the generated graph in GSR. We find that the model performance increases with the increment of $\epsilon$ and the optimal value is 0.95, while the performance decreases when $\epsilon$ goes beyond the optimal value because the graph will be too sparse with a larger simi-

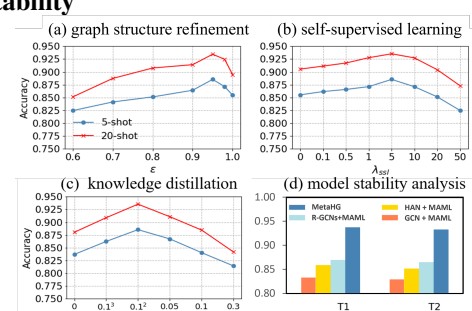

Figure 4: Hyper-parameter sensitivity (a)-(c) and model performance over different testing tasks (d).

larity threshold. Additionally, in Figure 4.(b), we vary the value of $\lambda_{ssl}$ in Eq. 7. By comparing with $\lambda_{ssl} = 0$ (without SSL) and $\lambda_{ssl} = 5$, we can further validate the effectiveness of SSL in enhancing the model performance. The optimal value of $\lambda_{ssl}$ is 5 and the performance drops with larger $\lambda_{ssl}$ due to the overfitting on the SSL task. Moreover, in Figure 4.(c), we vary the value of $\lambda_{kd}$ in Eq. 12 and the optimal value $\lambda_{kd}$ is 0.01 for our model, indicating that it is necessary to incorporate appropriate distilled knowledge. As shown in Fig. 4.(d), we also analyze the stability of our model by comparing with three baseline methods on different testing tasks (T1: opioids trafficker detection, T2: hidden trafficker detection). MetaHG is obviously more stable and also better than baseline methods on both T1 and T2, showing the robustness and effectiveness of MetaHG.

## 5.6 Embedding Visualization

To examine the effectiveness of our model intuitively, we visualize embeddings of drug traffickers generated by cFeature + DNN, R-GCNs + DNN, R-GCNs + MAML, and MetaHG respectively in Figure 5. The blue points represent the embeddings for opioids traffickers while grey points show the embeddings for regular users on Instagram. We can see that MetaHG generates the most distinct boundaries and the smallest overlapping area between opioids traffickers and regular users, demonstrating the superiority of our model.

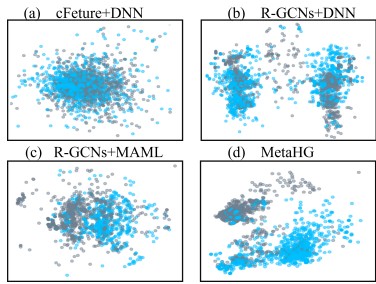

Figure 5: Embedding visualizations.

## 5.7 Case Study

We further analyze drug traffickers detected by MetaHG. Figure 6 illustrates the superiority of our model by comparing it with baseline models for real cases. MetaHG is able to detect hidden drug traffickers on social media although they pretend to be regular users or innocent users who seldom post any drug related information to the homepage. In this figure, drug trafficker "se***11" pretends to be an innocent user who does not have any illicit information on his homepage. However, he advertises at least four types of drugs (i.e., LSD, MDMA, DMT, and Shrooms) by leaving the contact information of encrypted chat tools (i.e., WICKR, Kik, and Snapchat) to other user's posts. These hidden drug traffickers cannot be detected by existing models, while MetaHG can detect these hidden drug traffickers based on the relationships among entities on social media.

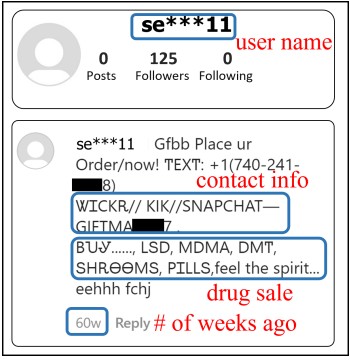

Figure 6: A hidden drug trafficker.

# 6 Conclusion

In this paper, to solve the problem of illicit drug trafficker detection with limited labeled data constraints, we create a new dataset and develop a novel system called MetaHG. Specifically, we first construct a HG to describe the structural relation and multi-modal content information in the dataset, and employ graph structure refinement to learn a refined HG. Then we employ R-GCNs with a self-supervised module to learn the node embedding in HG. Afterwards, we design a meta-learning algorithm to optimize the model. A knowledge distillation module is further introduced to improve model optimization. The extensive results demonstrate the effectiveness of our model by comparison with many baseline models.

# 7 Acknowledgment

This work is partially supported by the NSF under grants IIS-2107172, IIS-2140785, CNS-1940859, CNS-1814825, IIS-2027127, IIS-2040144, IIS-1951504 and OAC-1940855, the NIJ 2018-75-CX-0032.

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
