# Distilling Meta Knowledge on Heterogeneous Graph for Illicit Drug Trafficker Detection on Social Media - Supplementary Material

**Yiyue Qian**[1,2]**, Yiming Zhang**[1]**, Yanfang Ye**[1,2*]**, Chuxu Zhang**[3*]
[1]Department of Computer and Data Sciences, Case Western Reserve University, USA
[2]Department of Compute Science and Engineering, University of Notre Dame, USA
[3]Department of Computer Science, Brandeis University, USA
{yiyue.qian, yiming.zhang10, yanfang.ye}@case.edu, chuxuzhang@brandeis.edu

In the supplementary material, we first introduce the details of data preparation. To help reproduce our model and all baseline models, we provide a detailed description of experimental settings (including experimental environment, the value of hyper-parameters, iteration number, and details of reproducing baseline models). Additionally, we conduct additional experiments to validate the robustness of MetaHG and discuss the potential ethical issues as well as the limitation of our paper.

## 1 Dataset Details

### 1.1 Data Preparation

Based on the official Instagram APIs [2], we collect 8,651 users and 79,705 posts from Instagram. According to the drug types defined by National Institute on Drug Abuse [7], we manually classify these users into six groups (including five types of drug traffickers and regular users) based on the functions of drugs they post on Instagram (see Table 1). In Table 1, the left column represents the drug trafficker group and the right column shows some examples of the corresponding drugs belonging to the group. Regular users are those who are irrelevant to drug trafficking activities. For instance, hallucinogens traffickers are defined as those who advertise and sell hallucinogens-related drugs (e.g., LSD and DMT). Note that mixture traffickers are those who sell at least two groups of drugs. For instance, a drug trafficker who sells stimulants-related drugs (e.g., cocaine and methamphetamine) and opioids-related drugs (e.g., fentanyl and oxycodone) is defined as a mixture drug trafficker. In this paper, according to the number of labeled samples, we consider three types of drug traffickers (i.e., stimulants trafficker, hallucinogens trafficker, mixture trafficker) as meta-training task data and the rest of two types (i.e., opioids trafficker and hidden trafficker) as meta-testing task data.

Table 1: The different types of drug traffickers and their related drugs.

| Trafficker Type | Drugs |
|---|---|
| Stimulants trafficker | cocaine, meth (crystal meth), amphetamine, methamphetamine, weed |
| Hallucinogens trafficker | LSD, MDT, MDMA, ketamine, magic mushrooms, mescaline, hoasca |
| Opioids trafficker | oxycodone, hydrocodone, codeine, morphine, fentanyl, meperidine |
| Hidden trafficker | advertise drugs mostly by leaving the contact information to others' posts |
| Mixture trafficker | sell at least two different groups of drugs (e.g., cocaine, codeine, and LSD) |

---

[*]Corresponding authors.

35th Conference on Neural Information Processing Systems (NeurIPS 2021).

## 2 Experimental Details

### 2.1 Baseline Setting

We employ five sets of baseline models (twenty) in this paper. To compare traditional classifiers (B1, B2, and B4) with few-shot learning models fairly, we utilize all labeled data in meta-training tasks and few-shot labeled data (support set) in meta-testing tasks for model training, and then use the rest of the data in meta-testing tasks (query set) for model evaluation.

For B1 group, we take text feature (tFeature), image feature (iFeature), and the combination of text and image features (cFeature) as features vectors for users respectively, and then feed them into a generic 4-layer **DNN** [11] classifier to detect drug traffickers. For B2 group, we reproduce the method [5] by implementing a recurrent neural network with an LSTM unit to study the pattern of users with mere tFeature to detect drug traffickers. Additionally, we reproduce the model [6] by utilizing the biterm topic model with single tFeature to learn the latent patterns of users for drug traffickers detection. For B4 group, we perform **DeepWalk** [8] to learn node embedding (ignoring the heterogeneous property and attribute information) by modeling structure proximity. Besides, we use **metapath2vec** [1] to learn the semantic information of three defined meta-paths [12] in this application. (e.g., $user_1 \xrightarrow{have} post_1 \xrightarrow{tag} keyword \xrightarrow{tag^{-1}} post_2 \xrightarrow{have^{-1}} user_2$). In addition, we perform four graph neural network based representation learning models including **GCN** [4], **GAT** [13], **HAN** [15], and **R-GCNs** [9] to learn the node embedding in HG by leveraging both node features and graph structure information. In particular, for HAN, we utilize three defined meta-paths which are the same as metapath2vec and implement HAN to learn the attention-based node embedding. Similar to B1 and B2, we feed the learned user embedding to a 4-layer DNN classifier to detect illicit drug traffickers.

For few-shot learning methods in B3 and B5, we consider few-shot samples in each task as the support set for model training and the rest of data in each task as the query set for model evaluation. For B3, we implement three popular methods including **MAML** [3], **MatchingNet** [14], and **ProtoNet** [10] with cFeature for model optimization. Both MatchingNet and ProtoNet are metric learning based models and we utilize the cosine distance as the metric for both models in this application. MAML is a gradient-based model to learn well initialized model parameters that can be quickly adapted to new tasks and we set the base model of MAML as a 2-layer neural network. For B5, we feed the user embedding generated by six different graph representation learning models mentioned above to MAML for drug traffickers detection.

### 2.2 Evaluation Metrics and Parameter Settings

To evaluate the performances of our model and baseline methods, we adopt two widely-used metrics: accuracy (**ACC**) and F1 score (**F1**). We apply Pytorch to implement all methods and all experiments are conducted under the environment of the Ubuntu 16.04 OS, plus Intel i9-9900k CPU, GeForce GTX 2080 Ti Graphics Cards, and 64 GB of RAM. For the meta-learning model, inner-level and outer-level learning rates are set as 0.05 and 0.08 respectively. Additionally, the optimal hyperparameters of $\epsilon$, $\lambda_{ssl}$, and $\lambda_{kd}$ are 0.95, 5, and 0.01 respectively. For graph representation learning models, the dimension of node embedding is 200, and the iteration number is 200. We run 50 times for each experiment by changing the value of random seeds and then acquire the final average results.

### 2.3 Experimental Results

To validate the robustness and effectiveness of our model MetaHG, we further conduct two sets of experiments on different tasks T1 (opioids traffickers detection) and T2 (hidden traffickers detection) respectively. In Table 2, We can conclude that MetaHG is robust on both tasks and significantly outperforms other baseline models.

## 3 Discussion and Limitation

Relied on the official Instagram APIs, all information we collected is public and we never collect any private and personal user information. Additionally, all drug trafficker samples illustrated in this paper are anonymous and would not be harmful to the user. Therefore, we don't expect any privacy

Table 2: Performance of $\pm 95\%$ confidence intervals of different models on each type/task ($T_{id}$).

| Setting | | 5-shot | | 20-shot | |
|---|---|---|---|---|---|
| $T_{id}$ | Model | ACC | F1 | ACC | F1 |
| $T_1$ | GCN+MAML | 0.7864 ±0.010 | 0.7852 ±0.009 | 0.8334 ±0.010 | 0.8375 ±0.008 |
| | HAN+MAML | 0.8084 ±0.008 | 0.7986 ±0.008 | 0.8587 ±0.008 | 0.8353 ±0.007 |
| | R-GCNs+MAML | 0.8284 ±0.006 | 0..8175 ±0.005 | 0.8694 ±0.005 | 0.8562 ±0.005 |
| | **MetaHG** | **0.8884**±0.003 | **0.8773** ±0.002 | **0.9374** ±0.002 | **0.9341** ±0.003 |
| $T_2$ | GCN+MAML | 0.7814 ±0.010 | 0.7803 ±0.008 | 0.8294 ±0.010 | 0.8327 ±0.007 |
| | HAN+MAML | 0.8035 ±0.008 | 0.7924 ±0.008 | 0.8513 ±0.007 | 0.8301 ±0.007 |
| | R-GCNs+MAML | 0..8231 ±0.006 | 0.8123 ±0.005 | 0.8647 ±0.006 | 0.8504 ±0.005 |
| | **MetaHG** | **0.8842** ±0.003 | **0.8731** ±0.002 | **0.9328** ±0.002 | **0.9297** ±0.002 |

and ethical issues. Due to privacy regulations, our collected data are further processed including anonymizing some user information. As collecting data on social media consumes extensive energy, we collect the data on Instagram as a showcase to analyze drug traffickers on social media platforms. In the future, to effectively solve the drug trafficking problem, we wish to study these illegal activities on more social media platforms and further demonstrate the effectiveness of our model.