# OpenReview forum: "Distilling Meta Knowledge on Heterogeneous Graph for Illicit Drug Trafficker Detection on Social Media"
_NeurIPS.cc/2021/Conference — NeurIPS 2021 Poster_

### Official Review · Reviewer_VABt · 2021-07-10

**Rating:** 4
**Confidence:** 4

**Summary:**

Summary:
The authors propose a model to detect drug-trafficker profiles on Instagram. The model consists of many steps and phases, but in short: they create a heterogeneous graph from users, keywords, and posts, and in addition to the regular cross-entropy loss they propose to add multiple regularizer terms extracted from several stages of their algorithm (i.e., a term to build the graph, a term to learn the node representations, and a term to distill the knowledge from an out-of-domain model).

To evaluate their model, the authors created a dataset and compared their model with many baselines and their combinations. They also reported several experiments, including user visualization and a case study.

**Limitations And Societal Impact:**

They have addressed it.

**Main Review:**

Reasons to accept:

- The model works very well in the dataset. The improvement over the baselines is substantial.

- The authors release their code (not the dataset).


Reasons to reject:

- The model is enormous, it has many moving parts. This makes the model generalization to other datasets and scenarios very questionable. The sheer size of the model is not the main reason to reject, however this property cannot be evaluated with only one dataset (as the authors did). The astronomical performance of the model might be due to particular properties of this dataset—this is very likely, because the model is employing many aspects and areas.

- The model has many hyper-paremeters—I think five. None of them are discussed how they are tuned. This combined with the complexity of the model severely distorts the conclusions made in the paper. The last but not least is that the evaluations are made in the low-resource data regime. Practically such a gigantic model with many hyper-parameters is not tunable in semi-supervised settings.

- Even though the model has several steps, almost all of the employed techniques pre-exist. The actual contribution is very limited.

- Other minor issues:

  - The methodology section is not written well, it has several grammatical issues.

  - It is not described that where the numbers reported in Table 5 come from, what they represent, etc—in this table the main results are reported.

  - In Equation 4, are not X_i and X_j projected into the same space? Why do they have different weight vectors W_s1 and W_s2?

  - In Table 3, the ablation study shows that the meta-learning step is the most influential technique in the model. It is so effective that if we drop it from the model, the proposed model turns into a mediocre model, at best. Why is that? Do the baselines that employ meta-learning have the same module? This is not discussed in the paper.

**Time Spent Reviewing:**

6

---

> ### Author Response · Authors · 2021-08-08
> **Response to Reviewer VABt**
>
> We truly appreciate your valuable comments and we have made clarifications to your questions as follows.
>
> Q1: The model is enormous and it has many moving parts. The model cannot be evaluated with only one dataset.
>
> A1: Illicit drug trafficker detection on social media platforms is a very challenging problem. The major challenges include modeling multi-modal social media data, sparse social (graph) connection and limited labeled data of drug traffickers. Thus, we developed a novel model MetaHG which contains several complementary components to address these challenges and solve the problem collaboratively. MetaHG can be applied to datasets from different social media platforms (e.g., Instagram or Twitter) since they have similar data formats (e.g., the relational connections among users, post text/image information and user profile information). However, collecting and labeling datasets from social media platforms is time and resource consuming. In this paper, we collected and labeled the dataset from Instagram as a showcase. We (including 3 groups, 2 annotators in 288 each group) spent two months collecting this dataset and manually labeling users. The dataset will be shareable upon paper acceptance. Additionally, we plan to collect and label another dataset from Twitter.
>
>
> Q2: Too many hyper-parameters and no discussion about how to tune them.
>
> A2: We have discussed the sensitivity of key hyper-parameters and how to tune hyper-parameters in section 5.6 (Hyper-parameters Sensitivity and Model Stability) and section 3.2 (Evaluation Metrics and Parameter Setting) in the supplementary materials. Specifically, there are four key hyper-parameters in the joint learning model including one similarity hyper-parameter controlling the sparsity of feature similarity graph and three trade-off hyper-parameters balancing the self-supervised learning, graph structure learning and knowledge distillation respectively. For graph structure learning, $\epsilon$ in Equation 3 is the similarity threshold to control the density of feature similarity graph and we tuned the value of $\epsilon$ in the range of \{0.6, 0.7, 0.8, 0.9, 0.95, 0.98, 1.0\}. We found the optimal value of $\epsilon$ is 0.95 (Figure 4(a)). Additionally, we tuned $\lambda_{gsl}$ and $\lambda_{ssl}$ to balance two objectives in the joint objective function (Equation 7). The optimal value of $\lambda_{gsl}$ and $\lambda_{ssl}$ are 0.0001 and 5 respectively. Moreover, in the knowledge distillation module, we searched the value of $\lambda_{kd}$ in the range of \{0, 0.001, 0.01,0.05, 0.10, 0.30\} and found the optimal value is 0.01 (Figure 4(c)).
>
>
> Q3: The actual contribution is very limited.
>
> A3: There are several contributions of this work. (1) As far as we know, this is the first work proposing a deep graph learning model to solve the problem of Illicit drug trafficker detection on social media. The proposed model contains several complementary components to address key challenges collaboratively, which is novel. (2) The proposed model outperforms all existing baseline methods for solving the problem, which is significant. (3) We collected and labeled a new dataset from Instagram, which will be sharable after paper acceptance. The new dataset will be valuable for this research and future work. We believe this work can bring impact to public health and help build up a drug-free world.
>
>
> Q4: Grammatical issues.
>
> A4: We will carefully check and correct the grammatical issues.
>
>
> Q5:  Where do the numbers in Table 5 come from? What do they represent?
>
> A5: There is no table 5 in the paper. Could you help check it?
>
>
> Q6: Why does the Equation 4 have two weight vectors?
>
> A6: In Equation 4, $W_{s1}$ and $W_{s2}$ are weight parameters measuring the importance of different feature dimensions for $X_{i}^{\prime}$ and  $X_{j}^{\prime}$ respectively.
>
>
> Q7:  Why is meta-learning so effective? Whether do all baselines employ the same setting?
>
> A7: One major challenge of this problem is that there are only limited labels of drug traffickers in social media. Meta-learning (e.g., MAML [1]), as a typical technique to tackle limited (few-shot) labeled data, can generate good model initialization which can be fast adapted to new test data with limited labels. Thus, we were motivated to incorporate meta-learning into the deep graph learning model and obtained very good performance. For the fair comparison, all baseline methods employ the same data setting (few-shot) and we have discussed this in detail in section 3.1 (Baseline Setting) in the supplementary materials.
>
>
> We sincerely hope you can reconsider the score if our response addresses your concerns. Please let us know if you have any more questions.
>
>
> [1] Model-agnostic meta-learning for fast adaptation of deep networks, ICML 2017

---

> > ### Comment · Reviewer_VABt · 2021-08-24
> > **Re: Response to Reviewer VABt**
> >
> > Thank you for your detailed comment.
> >
> > ----------------
> >
> > I argued that the model is cumbersome, has many modules, and uses many hyper-parameters. This necessitates multiple datasets for evaluation, and requires fair and careful hyper-parameter tuning, otherwise this model can easily fit to any dataset but miserably fail to generalize. On the contrary to this argument, the authors in the paper have used only one dataset, and in doing so, they have evaluated the model in the low-resource data regime, where even simple models can easily overfit and result is astronomical accuracy.
> >
> > In response, the authors stated that the research task is difficult and they have spent 2 months to collect data.
> >
> > This is not a convincing argument. This paper claims that it is proposing a novel model for detecting drug traffickers. But my argument is that this claim is not validated in the paper. It is well-known that big models can easily overfit and require extensive evaluations, this paper ignores this simple fact.
> >
> > The argument that because this task is a difficult research problem and therefore we are allowed to glue together a large collection of techniques and avoid taking the steps to fully evaluate generalization is unacceptable. This is misleading.
> >
> > If the authors cannot fully evaluate their model yet, I suggest preparing a paper on the difficulties of detecting drug traffickers and publishing it in a related venue—not at NIPS.
> >
> > ----------------
> >
> > I stated that the authors have not discussed hyper-parameter tuning in the paper.
> >
> > In response, the authors stated that they have discussed this in Sections 5.6 and 3.2 (Appendix).
> >
> > This is incorrect. There is no such a discussion in the paper.
> >
> > The authors in their response begin describing the tuning procedure. Again, they don’t mention what is the validation set.
> >
> > This is unacceptable. It is unclear what has been really done during the experiments.
> >
> > ----------------
> >
> > I asked why after leaving out the meta-learning module, the method performance drops below some of the baselines. In fact, this “could” be a sign of model overfitting.
> >
> > The authors had no justification for it.
> >
> > ----------------
> >
> > I respect the reviewers’ opinion, but I will stand by the rejection of this paper.
> > Proposing a machine learning model without fully evaluating it, is like going to a dealer shop and buying a toy car—it cannot get you anywhere.
> >
> > P.S. I was referring to Table 2, in my initial review.

---

> > > ### Author Response · Authors · 2021-08-25
> > > **Additional Response to Reviewer VABt**
> > >
> > > Thanks for your additional comments and we make clarifications to your questions as follows.
> > >
> > > Q1: The model is cumbersome. The authors only used one dataset and evaluated the model in the low-resource data regime. This paper claims that it is proposing a novel model for detecting drug traffickers. But my argument is that this claim is not validated in the paper. It is well-known that big models can easily overfit and require extensive evaluations, this paper ignores this simple fact. If the authors cannot fully evaluate their model yet, I suggest preparing a paper on the difficulties of detecting drug traffickers and publishing it in a related venue — not at NIPS.
> > >
> > > A1: As we illustrated in the previous response, MetaHG is a novel graph learning model with several key modules to collectively address the challenges of multi-modal data, sparse social connection and limited labeled data which are shared across different social media platforms. To show effectiveness of different modules, sufficient ablations studies are provided in the paper (Section 5.4). As an early work for applying deep graph learning to drug trafficker detection on social media, we took Instagram as a showcase and conducted extensive experiments to fully evaluate MetaHG over different settings and tasks. In addition, the model can be easily applied to other social media platforms (e.g., Twitter) because data from these social media platforms share similar properties. In addition, we argue that our Instagram dataset is an important contribution of this work. As is known to all, NeurIPS encourages new machine learning dataset contribution (https://nips.cc/Conferences/2021/CallForDatasetsBenchmarks). We spent a lot of time/resource to create this new dataset which will be shareable upon paper publication. Additionally, we are collecting another dataset from Twitter. We believe our newly created data will be useful for future research. Moreover, NeurIPS welcomes submissions of all kinds of machine learning applications (https://nips.cc/Conferences/2021/CallForPapers). We believe our work which applies deep learning to combat drug traffickers is suitable for this conference.
> > >
> > >
> > > Q2: I stated that the authors have not discussed hyper-parameter tuning in the paper. There is no such a discussion in the paper. Again, they don’t mention what is the validation set. This is unacceptable. It is unclear what has been really done during the experiments.
> > >
> > > A2: We have described settings of hyper-parameters in the Section 3.2 of the supplement material. To be more specific, we applied grid search to tune the hyper-parameters. For example, we searched the value of $\lambda_{kd}$ (Eq. 12) in the range of {0, 0.001, 0.01,0.05, 0.10, 0.30}. We also did the hyper-parameter sensitivity analysis which can be found in Section 5.6. It provides the details of search ranges of different hyper-parameters. For data split, we set it as (80\%, 10\% and 10%) for model training, validation (tuning) and testing. We will add more experimental setting details (e.g., training, validation, and testing split) in the paper. In addition, we have shared the code of this paper for clear reference.
> > >
> > >
> > > Q3: I asked why after leaving out the meta-learning module, the method performance drops below some of the baselines. In fact, this “could” be a sign of model overfitting. The authors had no justification for it.
> > >
> > >
> > > A3: As we discussed in the previous comments, the reason is that meta-learning, a typical technique to tackle few-shot labeled data problem, can generate good model initialization which can be fast adapted to new test data with limited labels. Our model consistently outperforms all baseline methods in various settings. After removing meta-learning module, our model is better than all non-meta-learning learning methods while worse than some meta-learning methods (e.g., GCN + MAML). It further validates the effectiveness of meta-learning since these better baseline methods are equipped with this technique. We have provided detailed discussion in Section 5.3 and 5.4.
> > >
> > > Q4: I was referring to Table 2, in my initial review.
> > >
> > > A4: As shown in the Table 2, numbers (ACC and F1) represent model accuracy and F1 scores obtained in test data.

---

> > > > ### Comment · Reviewer_VABt · 2021-08-25
> > > > **Re: Additional Response to Reviewer VABt**
> > > >
> > > > I am not changing my verdict, because there is no evidence to show that this model can generalize to other datasets/scenarios. But I respect the decision made by the area chair, if they decide to go with accept.
> > > >
> > > > --------
> > > >
> > > > To the authors:
> > > >
> > > > Thanks for accepting to release your dataset. In the paper submitted to the conference you have mentioned:
> > > >
> > > > 	“Due to privacy regulations, the collected data will not be publicly accessible at this time ...”
> > > >
> > > > In case the paper gets accepted, please describe “in the paper” that what measures you take to preserve the user privacy.
> > > >
> > > >
> > > > Also in the section related to the hyper-parameters in Appendix, you have mentioned (which is the only description about tuning):
> > > >
> > > > 	“For the meta-learning model, inner-level and outer-level learning rates are set as 0.05 and 0.08 respectively. Additionally, the optimal hyperparameters of λssl, and λkd are 0.95, and 0.01 respectively.”
> > > >
> > > > Please replace this “in the paper” with the precise description of the tuning procedure, so that later researchers can reproduce your results.

---

> > > > > ### Author Response · Authors · 2021-08-26
> > > > > **Additional Response to Reviewer VABt**
> > > > >
> > > > > Thank you again for additional comments and we make clarifications to your questions as follows.
> > > > >
> > > > > Q1: There is no evidence to show that this model can generalize to other datasets/scenarios.
> > > > >
> > > > > A1: The point is that there is no open data for this important problem, thus we spent a lot of time/resource to create new dataset. It is an important contribution since the new dataset can help advance this research. In addition, as we mentioned in the previous response, extensive experiments have been conducted to demonstrate effectiveness of our model. In addition, the model can be easily applied to different social media platforms because their data share similar properties.
> > > > >
> > > > > Q2: In case the paper gets accepted, please describe “in the paper” that what measures you take to preserve the user privacy.
> > > > >
> > > > > A2: Thank you for suggestion. We will add description about how to preserve the user privacy for data sharing. For example, we will use hash tool to anonymize the username and profile information of users. We will also convert private information in posts (e.g., addresses, contact information) to hash values.
> > > > >
> > > > > Q3: Please replace this “in the paper” with the precise description of the tuning procedure, so that later researchers can reproduce your results.
> > > > >
> > > > > A3: In the previous response, we have discussed hyper-parameters tuning procedure and their sensitivity analysis. We will add more description to the paper if it is accepted. In addition, we have shared code for later researchers.

---

### Official Review · Reviewer_jGcf · 2021-07-16

**Rating:** 7
**Confidence:** 3

**Summary:**

This paper propose a novel framework MetaHG based on R-GCN for illicit drug trafficker detection on social media. A series of methods including meta learning(MAML), self-supervised learning(AS-SSL) and knowledge distillation(KD) are used to MetaHG, and obtain better metrics than traditional models with few labeled samples. In addition, the paper also creates a new dataset from Instagram for the task.

**Limitations And Societal Impact:**

None.

**Main Review:**

This paper combines a series of methods including R-GCN, meta learning, self-supervised learning and knowledge distillation for illicit drug trafficker detection, and obtain better metrics than traditional models. A new dataset is also created for this task, which is a valuable contribution. Paper writing is overall very clear and persuasive with adequate experiments and high performance.

Some questions for the author:
1. The graph model seems to be undirected, but social network relationships are obviously directed.
2. In Figure 4(d), what’s the definition of stability? How does it calculate and reflect stability?
3. In case study, the homepage of “se***11” writes out several drug names straightforwardly. Why can’t existing models detect it? What improvements help MetaHG detect it?

Some small points and feedback:
1. In Figure 2(a), the word “vendor” is not mentioned which should be “user”.
2. I think that creating a new dataset is valuable, the authors would consider mentioning it in the abstract.


**Time Spent Reviewing:**

6

---

> ### Author Response · Authors · 2021-08-08
> **Response to Reviewer jGcf**
>
> We truly appreciate your valuable comments and we answer your questions as follows.
>
> Q1: The graph model seems to be undirected?
>
> A1: Relationships on social media are directed. However, we thought that each connection can somewhat represent mutual correlation between users and using undirected connections can make HG denser. We have tried directed HG while found that the performance become worse due to relatively sparse graph connections. Thus, in this work we used undirected HG.
>
>
> Q2: In Figure 4(d), what's the definition of stability? How does it calculate?
>
> A2: In this paper, stability means the performance of models on different tasks is steady. We took two classification tasks as examples in Figure 4(d). T1 and T2 are two binary classification tasks of detecting two types of drug traffickers (e.g., hallucinogens trafficker, and opioids trafficker). As shown in Figure 4(d) in the paper and Table 2 in the supplementary materials, the performance of MetaHG on different tasks are stable and consistently outperforms baseline methods.
>
>
> Q3: Why can not existing models detect hidden drug traffickers?
>
> A3: To avoid being banned, some drug traffickers always invent new tactics to evade detection by pretending to be regular users on social media platforms. As shown in Figure 1(c) and Figure 5, these hidden drug traffickers rarely post drug descriptions or images on social media, but instead, they advertise drugs implicitly using slang and leave encrypted chat tool contact information through their comments to other users' posts. Existing models including NLP technology or image detection technology failed to detect these hidden drug traffickers by merely considering text or images. However, MetaHG can handle these hidden drug traffickers because it builds a HG by considering multi-modal features (e.g., text and image) and the social relationships among users.
>
>
> Q4: Mistakes in Figure 2(a).
>
> A4: We will fix the typo ('vendor' to 'user') in Figure 2(a).
>
>
> Q5: The contribution of creating a new dataset.
>
> A5: We will mention the dataset contribution in the abstract. The dataset will be shareable upon paper acceptance.

---

### Official Review · Reviewer_77r1 · 2021-07-16

**Rating:** 6
**Confidence:** 4

**Summary:**

This work studied the task of detecting online drug trafficking using social media data. In particular, the authors proposed to use graph convolutional networks to model various types of entities in social media and the relations between them. Also, a meta-learning algorithm is used for the model optimization.

**Limitations And Societal Impact:**

The limitations and potential negative societal impact of this work are addressed.

**Main Review:**

This work presented an interesting idea of using the GCN-based method to model the social media network for online drug trafficking detection. The detailed comments are listed below:

1. The novelty of the proposed model is relatively limited. Meta-HG is mostly applying existing techniques, e.g., R-GCN and meta-learning, to the application of drug trafficking detection. The major contribution may come from the combination of FSG and original HG, and AS-SSL.

2. HG construction is not clearly described. First, some details of the text feature vector generation are missing. For example, how is the username embedded in BERT? The authors said they concatenate the extracted features. However, if that is the case, then embedding of post/user may have different dimensions since the post length, and user information varies. How is this situation handled by the proposed graph convolutional network? Second, it is not clear how the authors use the image feature to represent post/user. One post may contain multiple images, and one user may post multiple posts.  How is this situation handled during the HG construction?

3. The proposed method models every entity in the social network in the graph, which raises the concern of the time/space complexity of the proposed model. It seems not practical to do this in a real-world situation, especially when the model tried to learn HG and FSG at the same time.

4. How the HG and FSG are combined? By unioning the edge set of HG and FSG?

5. The description of AS-SSL needs to be more clear about the terms such as node features, node attribute features, node attributes, node embeddings, node representations, and node pair features.

6. What are the "node features" mentioned in line 228? Are they node features obtained in the HG construction? If this is the case, then it does not make sense to use node attribute features (which I assume is obtained in the HG construction) again for the classification task because this will make the AS-SSL part decoupled from the computation graph. If node features refer to the node embeddings learned generated while training on HG, it is better to highlight that the cosine similarity calculation is still a network layer, though no parameters are learned. Just to make sure that readers understand that the AS-SLL is still in the computation graph.

7. Is AS-SSL applied only on the user nodes because, in Figure 2c, only the "vendor" icon is used there. If so, the authors should state it clearly in the paper.

8. The dataset collected is kind of "ideal" for the task because 37% of the users are drug traffickers. In practice, when the users want to increase the "recall" of detection of drug traffickers detection, they will increase the list of keywords which will inevitably bring "noise" into the graph. So reporting the model's performance in a less ideal setting would be interesting.


**Time Spent Reviewing:**

6

---

> ### Author Response · Authors · 2021-08-08
> **Response to Reviewer 77r1**
>
> We truly appreciate your valuable comments and we have made clarifications to your questions as follows.
>
> Q1: The novelty of model is limited.
>
> A1: Illicit drug trafficker detection on social media platforms is a very challenging problem. The major challenges include modeling multi-modal social media data, sparse social (graph) connection and limited labeled data of drug traffickers. Thus, we proposed a novel model MetaHG which develops several complementary components (based on graph neural network, meta-learning and knowledge distillation) to address these challenges and solve the problem collaboratively. As far as we know, this is the first model which applies deep graph learning and meta-learning to solve the problem of Illicit drug trafficker detection on social media.
>
>
> Q2: Some details about HG construction.
>
> A2: We have included more details about HG construction in section 4.1 (HG Construction on Social Media) and in section 1.2 (Feature and Relation) in the supplementary materials. Specifically, for enumerated attributes, we applied one-hot encoding to convert it to a binary feature vector and then we concatenate all text feature vectors as the text feature for each node. Here username is considered as an enumerated feature and we converted it to a binary embedding vector.  Additionally, as introduced in Equation 2, we employed a mapping layer to project all of the node's features $X$ to the same $d_{c}$-dimensional space. Moreover, as described in the Supplementary Materials, we implemented $VGG19$ [1] to get the image embedding of each image and then converted the image embedding to a vector with $200$-dimension by PCA. If one post has more than one image, we will calculate the post image embedding by averaging all of the image embeddings. On the other hand, if one user has more than one post, then each post will be considered as a post node in the graph. After getting the image embedding and text embedding for each node, we concatenate the text embedding and image embedding as the input feature of each node.
>
>
> Q3: Time/space complexity.
>
> A3: The time complexity and space complexity of our model are comparable to baseline methods. We take some baseline methods as examples. R-GCN with MAML takes 108 minutes to train the model while HAN with MAML costs 146 minutes. For our model MetaHG, it costs 155 minutes to train the model. Comparing to baseline methods, MetaHG needs a little more time for graph structure learning. Similar to the time complexity, the memory usage (space complexity) of MetaHG is also comparable to baseline models. R-GCN with MAML takes 18 GB RAM while HAN with MAML takes 60 GB RAM. Our model MetaHG needs 40 GB RAM to train the model.
>
>
> Q4: How to combine the HG and FSG?
>
> A4: We combined the edges set of the original HG with the edges set of the generated feature similarity graph by union operation.
>
>
> Q5: The description of AS-SSL needs to be more clear about the terms.
>
> A5: Node features, node attribute features and node attributes are considered as the original features of nodes, while node embeddings and node representations are the node embedding in graph learning model. Additionally, node pair features are the original features pair of a node pair. We will use clear terms when describing these contents in the paper.
>
>
> Q6: More explanations of AS-SSL.
>
> A6: \'node features\' mentioned in line 229 is the original node features in HG. In this work, the self-supervised learning module is not a simple cosine similarity calculation while it is a learnable module to capture more graph information using unlabeled data and thus to enhance the representation learning ability. The input of the network is the concatenated node feature pairs and the output is the probability that node pairs are similar. The label of node pairs is discussed in section 4.3. The label of a node pair is 1 if the cosine similarity of the node feature pair is in the top-M list, otherwise 0. We will add more illustrations about this part to make it clear.
>
>
> Q7: Is AS-SSL only applied to user nodes?
>
> A7: Yes, the AS-SSL is only applied on the user nodes in HG. We will state it clearly in the paper.
>
>
> Q8: The dataset collected is kind of "ideal" for the task because 37\% of the users are drug traffickers.
>
> A8: The percentage of drug traffickers among all users in original collected data is small. In order to avoid the extremely imbalanced data issue and train the model stably, we selected a subset of original data with a relatively balanced ratio of regular users and drug traffickers. For fair comparison, this setting is applied to all methods. In addition, we have tried the imbalanced dataset (percentage of drug traffickers is relatively small) by adding more regular users in model training, our model still outperforms all baseline methods.
>
>
> We sincerely hope you can reconsider the score if our response addresses your concerns. Please let us know if you have any more questions.
>
>
> [1] Very deep convolutional networks for large-scale image recognition, ICLR 2015

---

> > ### Comment · Reviewer_77r1 · 2021-08-23
> > **Rating adjusted**
> >
> > Thanks for the response, and it did address most of my concerns.

---

> > > ### Author Response · Authors · 2021-08-25
> > > **Rating adjusted**
> > >
> > > Thank you for kind consideration. We really appreciate it.

---

### Official Review · Reviewer_igwA · 2021-07-17

**Rating:** 8
**Confidence:** 3

**Summary:**

This paper presents a novel heterogeneous graph learning model for automatically detecting illicit drug trafﬁckers on Instagram. The proposed model addresses two challenges: 1) sparse graph structure, and 2) limited labeled samples for model training. The authors also collected a large social media dataset from Instagram and conducted a lot of experiments to evaluate the proposed model.

**Limitations And Societal Impact:**

Yes.

**Main Review:**

Overall, this work provides a complete study with sufficient motivation, novel techniques, and thorough experiments. More speciﬁcally, the authors consider the constraints of graph sparsity and limited labeled data for model design, which includes the three-fold contribution: 1) a novel framework named MetaHG to integrate both structural relations and unstructured content information for illicit drug trafﬁcker detection. 2) graph structure learning is used to refine HG structure and self-supervised R-GCN is employed to learn the node embeddings. 3) a meta-learning approach with knowledge distillation is adopted to optimize the model which could overcome the issue of insufficient labeled samples. I think it is an interesting work and has the ability for generalizing to other applications.

On the other hand, the design of the method is reasonable and comprehensive. Two challenges of sparse graph connection and insufficient labeled samples are well addressed. Moreover, the presentation is good with straightforward and clear writing.

There are some points where improvements could help further strengthen the paper:
In the methodology section, the motivation of modeling HG is not very clear. Why the heterogeneous nodes and relations can be useful for this task?

The paper uses eight types of relations to describe the relationships among different types of nodes. The authors should explain how to select these relations. Do they all benefit from this task?

The proposed self-supervised augmentation model is mainly based on the content information. I'm curious about the performance if it leverages some relations to implement the same model.

Do the authors evaluate the training and testing time for the proposed model?


**Time Spent Reviewing:**

1.5 hours

---

> ### Author Response · Authors · 2021-08-08
> **Response to Reviewer igwA**
>
> We truly appreciate your valuable comments and we answer your questions as follows.
>
> Q1: Why are heterogeneous nodes and relations useful for this task?
>
> A1: As shown in Figure 1(c), some drug traffickers always invent new tactics to evade detection. They rarely post drug descriptions or images on social media, but instead, they advertise drugs implicitly using slang and leave encrypted chat tool contact information through their comments to other users’ posts. Most of the existing works based on a single type of content feature (e.g., text or image) fail to detect drug traffickers on social media platforms. Therefore, we considered multi-modal features (e.g., both text and image) and the structural relationships among entities (e.g., users and posts). These foxy drug traffickers can be detected by considering the text as well as image and relationships among users and posts on social media platforms.
>
>
> Q2: How to select these relationships among nodes? Are they useful for the detection problem?
>
> A2: At first, we defined twelve types of relationships among nodes (user, post, and keyword) based on the semantic relationships between nodes. Then we selected eight types of relationships by considering the performance of each relationship. We built an HG based on each relationship and then trained our model MetaHG. Afterwards, we picked up the top eight relationships to build the HG and then trained MetaHG to detect drug traffickers.
>
>
> Q3: Some relations can be added to the self-supervised augmentation.
>
> A3: We have tried to add relations among nodes to augment the representation learning model. However, we found the performance was not improved. We believed node features have some contributions to the representation learning and the performance of self-supervised augmentation based on node features was improved. Thus, the proposed self-supervised module is mainly based on the content information.
>
>
> Q4: Do authors evaluate the training time of MetaHG?
>
> A4: MetaHG costs 155 minutes to train the model. The training time of MetaHG is comparable to baseline methods.

---

### Official Review · Reviewer_Ykxj · 2021-07-18

**Rating:** 8
**Confidence:** 4

**Summary:**

The paper proposes a novel model MetaHG to automatically detect illicit drug traffickers on social media. It firstly builds a heterogeneous graph based on post content and relational structure information on social media to characterize the drug trafficking system. Then it integrates graph structure learning and relational-based GCN to learn robust node embedding on HG. Afterwards, it leverages meta-learning and knowledge distillation to optimize model parameters and further to detect drug traffickers on social media. Finally, it conducts extensive experiments to validate the effectiveness of the framework.

**Limitations And Societal Impact:**

This paper discusses the limitations and potential impact in the introduction and the supplement materials.

**Main Review:**

It is very interesting and meaningful to develop a novel model for drug trafficking detection on social media. I list the main strengths and weaknesses of the paper below.

Strengths:

+ This paper is well organized and easy to follow. It elaborates on background, related work, methodology, and experiments.

+ To address the few labeled data issue, the proposed framework leverages meta-learning and knowledge distillation to optimize the model. In addition, to solve the problem of graph sparsity, it combines graph structure learning and relational GCN augmented by self-supervised learning to learn the robust node embedding on heterogeneous graph. The design of MetaHG framework is novel the technique quality of this paper is solid.

+ The paper conducts comprehensive experiments to validate the effectiveness of the framework. It compares with five groups of baseline models, especially previous methods of the same topic. To show the effectiveness and robustness of the framework, it conducts detailed ablation studies for analyzing the effectiveness of each module and it does comparison experiments on different tasks for analyzing the model robustness.

Weaknesses:

- The paper doesn’t explain why the heterogeneous graph is sparse. It is common that the relationships among users and posts on social media platform (e.g., Instagram) can be dense. Why the heterogeneous graph in this paper is sparse?

- The description of baseline models and their settings is short and some parts are confused. In Table 2 Group B2, it shows the results of other two papers with different shots, which is confused. The paper should introduce these two methods briefly.

- Minor issue: In Figure 1 and Figure 5, I think “63w” and “60w” should be how many weeks ago did the comment post instead of the number of like. Please double-check it.

**Time Spent Reviewing:**

3h

---

> ### Author Response · Authors · 2021-08-08
> **Response to Reviewer Ykxj**
>
> We truly appreciate your valuable comments and we answer your questions as follows.
>
> Q1: Why is the heterogeneous graph built on social media sparse?
>
> A1: As discussed in section 1 (Introduction) and section 4.2 (HG Structure Refinement via Metric Learning), some users (e.g., skilled drug traffickers) on social media pretend to be inactive on social media to avoid being banned or being suspected. They rarely have posts or descriptions on their homepage but leave some advertisements to others' posts. So the relationship among these types of users, post, and keyword are sparse.
>
>
> Q2: The description of the baseline setting is short.
>
> A2: We have discussed and introduced the baseline models and their settings in section 5.2 (Baseline Methods) and section 3.1 (Baseline Setting) in detail. For the baseline models of group B2, we have introduced them in the supplementary materials.
>
>
> Q3: Minor issues in Figures.
>
> A3: 'w' in Figure 1 and Figure 5 denotes how many weeks ago did the comment post. We will correct them in the paper.

---

### Decision · Program_Chairs · 2021-09-27

**Decision:**

Accept (Poster)

**Comment:**

The reviewers generally agree that the paper is interesting and like the new application with a novel dataset that will be published, thorough experiments that establish a baseline, and a clear presentation. The reviewers provided recommendations that should clearly be incorporated in the paper.